# An Efficient Method for Document Correction Based on Checkerboard Calibration Pattern

Mina Ibrahim [1,*], Marian Wagdy [2], Fahd S. AlHarithi [3], Abdulrahman M. Qahtani [3], Wail S. Elkilani [4] and Sameh Zarif [1]

1 Department of Information Technology, Faculty of Computers and Information, Menoufia University, Shibin el Kom 6131567, Egypt
2 Faculty of Computer Science and Information Technology, Ahram Canadian University, 6th of October City 3221405, Egypt
3 Department of Computer Science, College of Computers and Information Technology, Taif University, Taif 26571, Saudi Arabia
4 College of Applied Computer Science, King Saud University, Riyadh 11451, Saudi Arabia
* Correspondence: mina.ibrahim@ci.menofia.edu.eg

**Abstract:** Portable digital devices such as PDAs and camera phones are the easiest and most widely used methods to preserve and collect information. Capturing a document image using this method always has warping issues, especially when capturing pages from a book and rolled-up documents. In this article, we propose an effective method to correct the warping of the captured document image. The proposed method uses a checkerboard calibration pattern to calculate the world and image points. A radial distortion algorithm is used to handle the warping problem based on the computed image and world points. The proposed method obtained an error rate of 3% using a document de-warping dataset (CBDAR 2007). The proposed method achieved a high level of quality compared with other previous methods. Our method fixes the problem of warping in document images acquired with different levels of complexity, such as poor lighting, low quality, and different layouts.

**Keywords:** de-warping document image; document correction; camera calibration; checkboard calibration pattern; radial distortion

## 1. Introduction

Paper documents are the most common type of document utilized in our daily lives. The most common method for retaining information from paper is to capture it. Photos of documents are easily retrieved and archived at any moment. We can transmit this image of the document to anyone in the world utilizing social media tools. However, many captured document images include issues such as perspectival and geometric abnormalities. When dealing with rolled documents or thick books, warping and curled lines are the most common problems that appear [1,2]. Figure 1 depicts warped document images taken from a bound book.

The technique of geometrically transforming 2D images is referred to as the 'document-image warping problem.' However, the term "warp" may appear to imply radial distortion. The term "document-image warping" encompasses a wide range of transformations, from complicated asymmetrical warps to basic transformations like rotation and scaling. The use of digital image de-warping has a wide range of applications. The common applications of image processing involve at least one image rotation, scaling, or translation, which are simple warping examples. Image de-warping is increasingly used in the field of remote sensing. Many geometric distortions in images acquired from various satellites, sensors, or aircraft are created by the curvature of the Earth or camera lenses. These geometric and lens distortions are effectively corrected via image de-warping. Image de-warping has recently gotten significant attention because of a special effects method called 'morphing'.

In this study, we look at how to fix the warping problem in camera-captured document images. Optical character recognition (OCR) utilizes the image of the acquired document to accomplish tasks like text extraction and character recognition. The de-warping technique is responsible for the majority of OCR systems' accuracy [3].

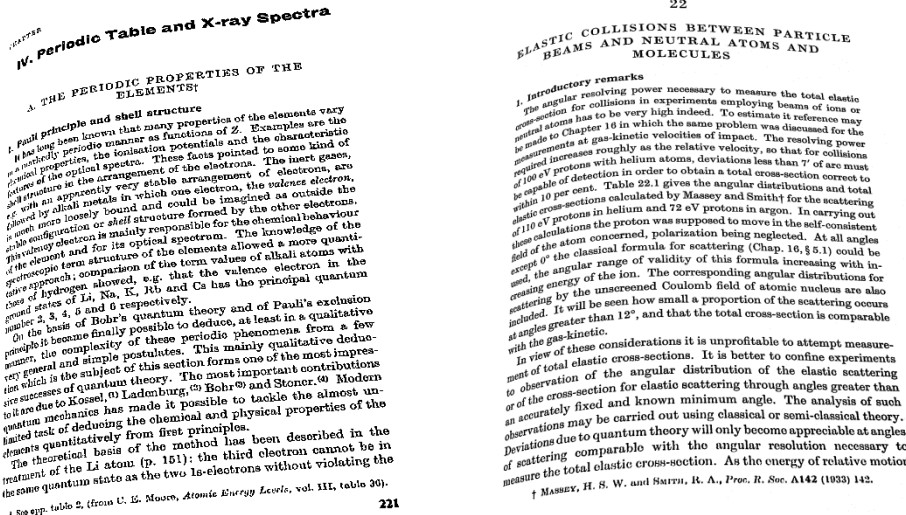

**Figure 1.** Example of document-image warping.

In this research, we present a document image de-warping approach based on a checkerboard calibration pattern. To compute world and image points, the first step is to apply well-defined checkerboard patterns. The world and picture points obtained in the first step are used in the second step to determine camera calibration parameters. Finally, the document image is de-warped using the derived camera-calibration parameters. The main contributions of this paper are as follows:

- We propose a novel document-image de-warping method based on a checkerboard calibration pattern.
- Well-defined checkerboard patterns are employed to compute world and image points.
- Calculated camera calibration parameters are employed to de-warp the document image.

The remainder of this work is arranged in the following manner. The state-of-the-art algorithms are discussed in Section 2. The proposed algorithm is discussed in Section 3. Section 4 analyzes and discusses the results of the proposed method. Finally, in Section 5, a conclusion is derived.

## 2. Related Work

De-warping approaches for document images have received much attention in the scientific community in recent years, and various approaches have been presented. Three types of approaches are available: text-based, shape-based, and deep-learning-based.

The first category depends on extracting features from text lines, words, and characters. The ridge-based coupled-snakes model was employed among the methods described in [4,5]. These authors used the coupled-snakes model to calculate the baseline, then determined the starting location of each text line. Finally, the perspective distortion was removed using a four-point homography technique. Schneider et al. [6] developed a technique for introducing a vector field from the generated warping mesh based on local orientation data. The image is fixed by using numerous linear projections to approximate the nonlinear distortion. By using a baseline detector, they were able to extract the needed local orientation characteristics.

Kim et al. [7] introduced a method based on discrete representations of text blocks and text lines, which are sets of connected components. Each text line's features were retrieved in this study. Using the Levenberg–Marquadt technique, a cost function was created to

solve the warping. To overcome the de-warping problem, a few approaches based on word information have been presented. B. Gatos et al. [8] have suggested breaking the document into words. The orientation and orientation of each word are determined. Finally, to correct the problem of warping, each word is rotated.

A curled baseline pair was utilized by Bukhari et al. [4]. Each corresponding top and bottom baseline pair's map characters are computed. The restoration procedure was carried out by Zhang et al. [9] using segmentation and thin plate splines. The text baselines and vertical stroke boundaries for each line were determined by Lu et al. [10]. Bolelli et al. [11] presented a method that covered each letter with a quadrilateral cell and then computed the center of each letter to adjust its orientation, resulting in a flat word.

The most common methods are the text-information based. They work well if the segmentation method is correct. On the other hand, these systems have the disadvantage of being susceptible to high degrees of curl and varying of line spacing. They are also sensitive to the image resolution and words used. Text-based algorithms are time-consuming and fail when dealing with complicated layouts. They also failed when it comes to documents with graphics and tables in them. These approaches are susceptible to a variety of distortions and result in a high number of segmentation mistakes.

The second category is shape recovery. These methods are based on recovering the shape of document. As shown in [12], these techniques rely on 2D-form recovery employing a curved surface-projection map to restore a rectangular 2D region. The uniform parameterization procedure in [13] is guided by a physical pattern. The authors of [14] use a technique that separates the page into three vertical sections and projects each one separately. The approaches presented in [15,16] use text lines to establish the curved shape, then use a surrounding transformation model to flatten it.

The concept behind 3D reconstruction is to turn a coiled paper into three dimensions (x, y, and z). De-warping distortion may be addressed by first determining the depth (z) dimension and then fitting the curved form [17–20]. Laser range scans [21], stereo cameras, eye scanners [22], controlled lighting [23], and computed tomography (CT) scans for opaque objects [18,24,25] are examples of specialized equipment needed in this procedure. In order to acquire the 3D form of the coiled document, previous knowledge is also ecessary.

The majority of 3D methods do not rely on the document's content. As a result, these approaches are more precise in representing the shape than 2D methods. In the event of complicated deformities like folds and rumples, 3D-based approaches are also accurate. 3D-based solutions, on the other hand, need a unique setup to record the 3D geometry, as well as more intricate hardware setups and more expensive hardware.

Other methods of reconstructing a document's surface form using shading techniques have been developed. The notion of form recovery using shading techniques is based on the use of varied illuminations to recover shape, which is referred to as shading. Zhang et al. [26–32] provided methods for creating a 3D model using the shading approach. To recover the 3D or 2D form of a document, shading approaches do not require specialist hardware. The fundamental disadvantage of these approaches, on the other hand, is that they fail when the lighting is poor.

Deep-learning-based approaches are the last category. Bukhari et al. [33] suggested a novel deep-learning-based technique. To overcome the de-warping problem, this solution uses an image-to-image translation deep-learning approach. It uses the deformed input picture and the clean image to transfer the image from one domain to another. This approach utilizes the pix2pixhd network [34], which uses GANs to achieve a high resolution The Pix2pix translation network with CGAN [34] is unable to create $256 \times 256$ images. In the case of documents including tables and figures, deep-learning-based algorithms function effectively [35]. However, they need a large quantity of data and costly GPUs. A rectifying distorted document framework is presented in [36]. A fully convolutional network (FCN) is used to estimate pixel-wise displacements. After that, pixel-wise displacements are used to rectify the document image. In [37], a learned query embedding set is adopted to

capture the global context of the document image. The geometric distortion is corrected by a self-attention mechanism and it decodes the pixel-wise displacement solution.

## 3. Proposed Method

It has been established that 3D page-shape reconstruction methods produce high-quality results when recovering the shape of the document regardless of the content of the document image. However, specialized and expensive hardware is required. Furthermore, prior knowledge parameters from the camera and scanner are needed. In contrast, 2D image-processing techniques do not require any parameters or hardware. Unfortunately, in the case of complex layouts and bad illumination, the shape is often inaccurately recovered. The proposed method combines the advantages of 3D and 2D methods by recovering documents with high quality regardless of their content and without prior knowledge parameters. To obtain the world and image points, the proposed method utilizes checkerboard calibration patterns. Regardless of the content of the document image, camera calibration parameters are produced depending on the computed world and image points. Finally, the radial distortion algorithm is used to correct the warping problems based on the extracted camera-calibration parameters. The block diagram of our method is illustrated in Figure 2.

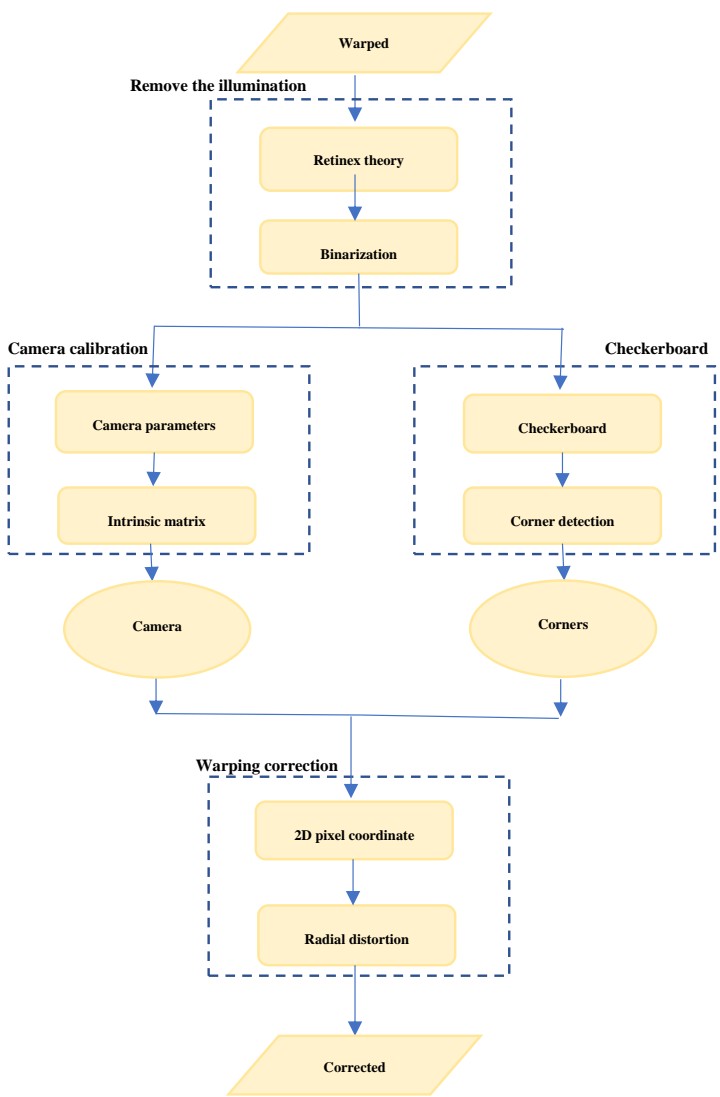

**Figure 2.** General block diagram of our method.

There are five main steps in the proposed technique. The first step includes removing bad illumination and shading from the document image. A well-defined checkerboard calibration pattern is used in the second step to match the warped shape of the document. Checkerboard corner detection is used in the third step to extract the world points. Camera calibration parameters are calculated in the fourth step. The calculated camera calibration parameters and the world points are used to de-warp the document image in the fifth step. The following subsections go through each stage in depth.

### 3.1. Remove the Illuminations

When capturing a page of a thick book, the resultant document image has shading in the spine area of the bound volume as shown in Figure 3a. OCR engines and segmentation algorithms face numerous challenges in recognizing the text in the case of bad illumination. The proposed method in this paper overcomes this issue by employing the retinex theory introduced in [38]. The image is mathematically represented in retinex theory as a function of its illumination $U(x, y)$ and the reflectance $F(x, y)$,

$$I(x, y) = U(x, y).F(x, y) \tag{1}$$

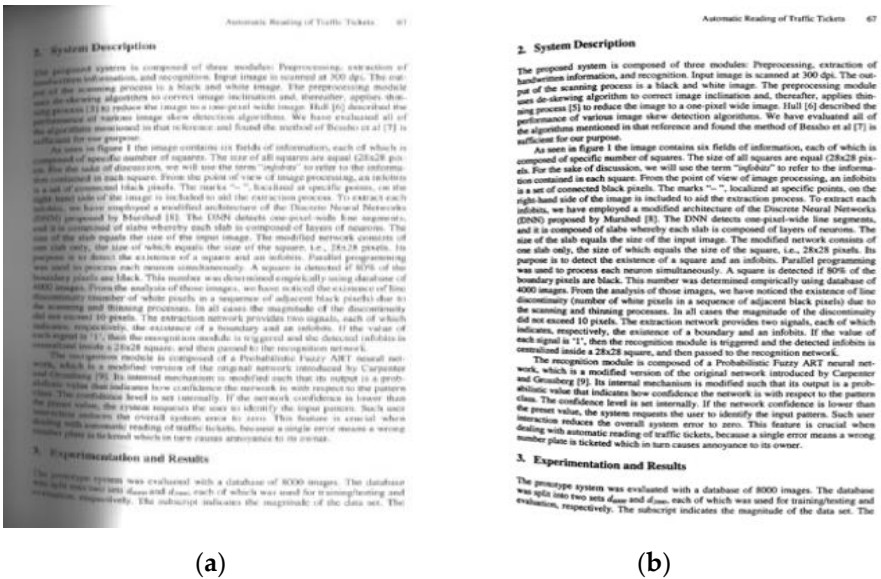

(**a**)            (**b**)

**Figure 3.** Two images of the same document before (**a**) and after (**b**) illumination removal.

The low-frequency component of the measured image can be used to approximate the illumination $U(x, y)$ component.

Lightness refers to the estimation of reflectance $F(x, y)$ and is calculated as:

$$F(x, y) = \frac{I(x, y)}{U(x, y)} \tag{2}$$

Binarizing the document image is simple after removing the illumination and lightness using the global binarization method described in [39]. A document image before and after the illumination-removal process is shown in Figure 3.

### 3.2. Checkerboard Calibration Pattern

In this step, the proposed method recovers the 3D points of the document image (world point) by using a well-defined calibration pattern. In the field of computer vision, different calibration patterns are used to perform numerous tasks, such as defining a set of markers, calculating camera parameters (focal length and center), and undistorting the

image. Figure 4 shows an example of well-defined calibration pattern images that are commonly used in computer vision. In this work, we use the last checkerboard calibration pattern to recover the 3D world points of the document image, by defining its corners as a set of markers which will be used in the calibration algorithm in Section 3.4.

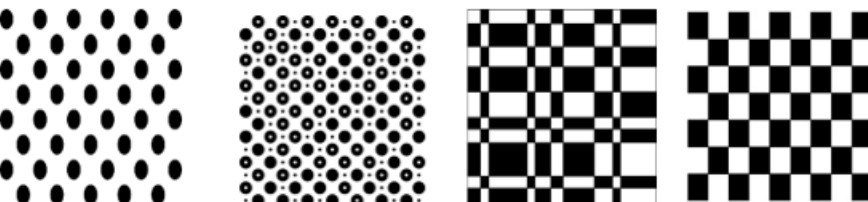

**Figure 4.** Example of a well-defined calibration pattern.

The proposed method collects different de-warped checkerboard patterns as shown in Figure 5. We manually select the corresponding checkerboard pattern mask, which has high similarity measures as in Figure 6. The size of the checkerboard must be at least six-by-six equal squares. The quality of the proposed method depends on the number of squares; a large number of squares produces a high quality image. The size of checkerboard has the same size of the corresponding document image.

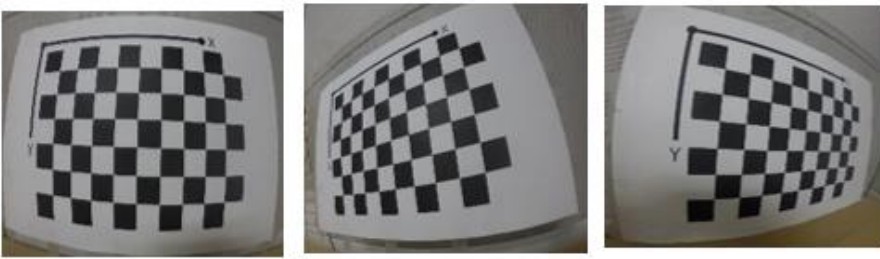

**Figure 5.** Example images of a checkerboard calibration patterns.

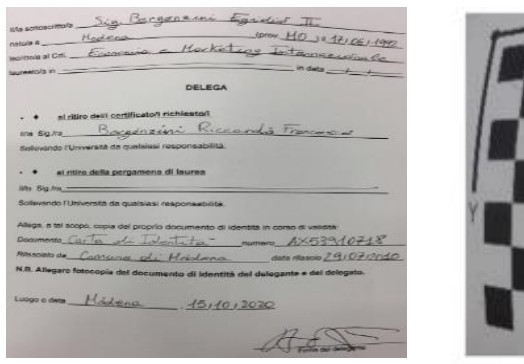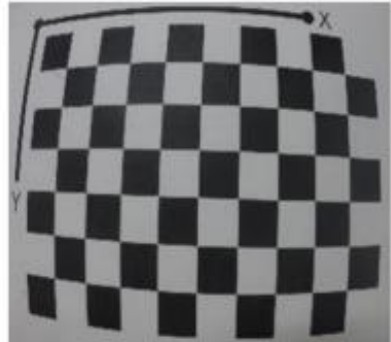

**Figure 6.** Example of selecting the corresponding checkerboard pattern.

### 3.3. Checkerboard Corner Detection

To simulate the 3D shape of the document image, we define the corresponding 3D world points $P_w (x, y, z, 1)$ of the checkerboard pattern. A set of markers are used to recover the shape by estimating the corners of the checkerboard. The checkerboard corners are extracted from each pair of black or white squares that touch each other. For each local corner located in the checkerboard, there are four squares surrounding the corner (upper left $U_l$, upper right $U_r$, lower left $L_l$, lower right $L_r$) as illustrated in Figure 7. The corner is selected if it satisfies the following condition,

$$U_l = L_r = black \text{ and } L_l = U_r = white \tag{3}$$

or

$$U_l = L_r = white \text{ and } L_l = U_r = black. \tag{4}$$

**Figure 7.** Local corner selection.

All points on the checkerboard lie in $[x\ y]$ plane; we drop the $z$ component $(z = 0)$ so the world points $P_w$ return an N-by-2 matrix of an N number of $[x\ y]$ coordinates of checkerboard corners. $P_w$ (0, 0) corresponds to the lower-right corner of the upper-left square of the board as shown in Figure 8a. World points $P_w$ are used to match and obtain the corresponding points from the warped document image $D_w$ to recover the 3D shape of the warped document image as shown in Figure 8b. Detected checkerboard corners as illustrated in Figure 8a reflect exactly the warped shape of the document image in Figure 8b. The matrix of $D_w$ is used to solve the warping problem of the document image as follows in the next subsection.

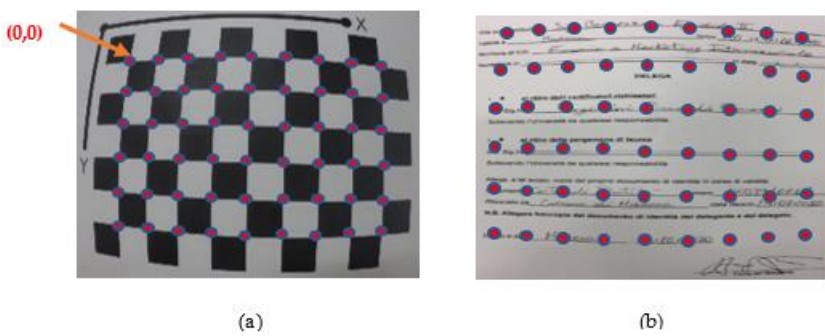

(a)                    (b)

**Figure 8.** Detecting the corners of checkerboard calibrating pattern in (**a**) and its corresponding warped document in (**b**).

*3.4. Camera Calibration*

To convert from a 3D world point to a 2D pixel coordinate, camera calibration is used to calculate the camera matrix (image points) [40–43]. Firstly, we must understand the camera-calibration process, which depends on two components: the intrinsic and extrinsic parameters of a camera. Extrinsic parameters are used to convert the document image into 3D camera coordinates (3D world point $P_w$) unlike the intrinsic parameters, which are used to convert the 3D camera coordinate into a 2D pixel coordinate. Figure 9 explains the difference between the intrinsic and extrinsic parameters.

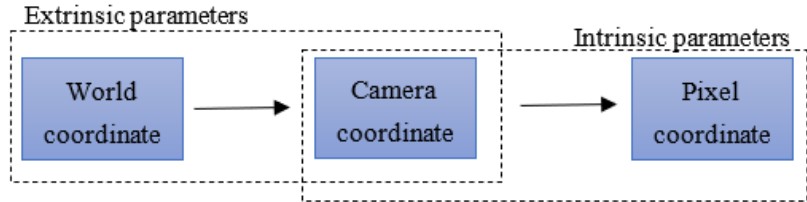

**Figure 9.** Intrinsic and extrinsic parameters.

Mathematically, the 3 × 4 camera matrix $P$ is defined as the following:

$$P = K\,[R\,|\,t] \tag{5}$$

Such that $K$ is the intrinsic camera matrix which describes the camera parameters. $[R \mid t]$ are the extrinsic parameters which consist of rotation $R$ and translation $t$.

The $3 \times 3$ intrinsic camera matrix $K$ is defined as

$$K = \begin{bmatrix} f & 0 & x_c \\ 0 & f & y_c \\ 0 & 0 & 1 \end{bmatrix} \tag{6}$$

where $f$ is the focal length and $(x_c, y_c)$ is the camera center. Unfortunately, the intrinsic camera parameters are unknown in the case of a document image, so we compute the camera parameters based on the following equations.

From the corresponding checkerboard pattern, the camera center $(x_c, y_c)$ is defined as

$$x_c, y_c = (W/2, H/2) \tag{7}$$

The focal length $f$ is defined as

$$f = \frac{W}{2} \left[ \tan \frac{\theta}{2} \right]^{-1} \tag{8}$$

such that $H$ is the height and $W$ is the width of the checkerboard pattern. $\theta = 180°$ is the field of view.

The intrinsic camera $K$ can be used directly to map from a 3D world coordinate $D_w$ of the document image to a 2D pixel coordinate $x_p (x_s, y_s)$, as the following,

$$x_p(x_s, y_s) = K D_w \tag{9}$$

where $k$ is the intrinsic camera matrix defined in Equation (5) and $D_w$ is a 3D world point of the document image as described in Section 3.3.

*3.5. Warping Correction*

Finally, the proposed method solves the warping problem of the 2D pixel coordinates $x_p(x_s, y_s)$ of the document image by using the radial distortion algorithm in [44–47], which depends on low-order polynomials, as shown as follows:

$$x_{dewarp} = x_s \left( 1 + R_1 r^2 + R_2 r^4 + R_3 r^6 \right) \tag{10}$$

$$y_{dewarp} = y_s \left( 1 + R_1 r^2 + R_2 r^4 + R_3 r^6 \right) \tag{11}$$

where $r^2 = x_s{}^2 + y_s{}^2$, $(x_s, y_s)$ is defined in Equation (8) and $(R_1, R_2, R_3)$ are the radial distortion parameters.

## 4. Results

Our method was evaluated on a workstation Core i5 2.5 GHz CPU and implemented in MATLAB R2020a. The performance of our algorithm was experimentally tested on some samples of warped document images. These samples have different levels of complexity, including, for example, degradation, handwriting, different languages, and different layouts. Figure 10 illustrates the result of the document image before and after warping correction. Figure 10a depicts the original document image, which it has numerous degradations, challenges, warping, and handwritten text. The document image after solving the degradation and warping problems is illustrated in Figure 10b. The corresponding checkerboard calibration pattern which was used is introduced in Figure 10c. Figure 10d presents the checkerboard after the de-warping process.

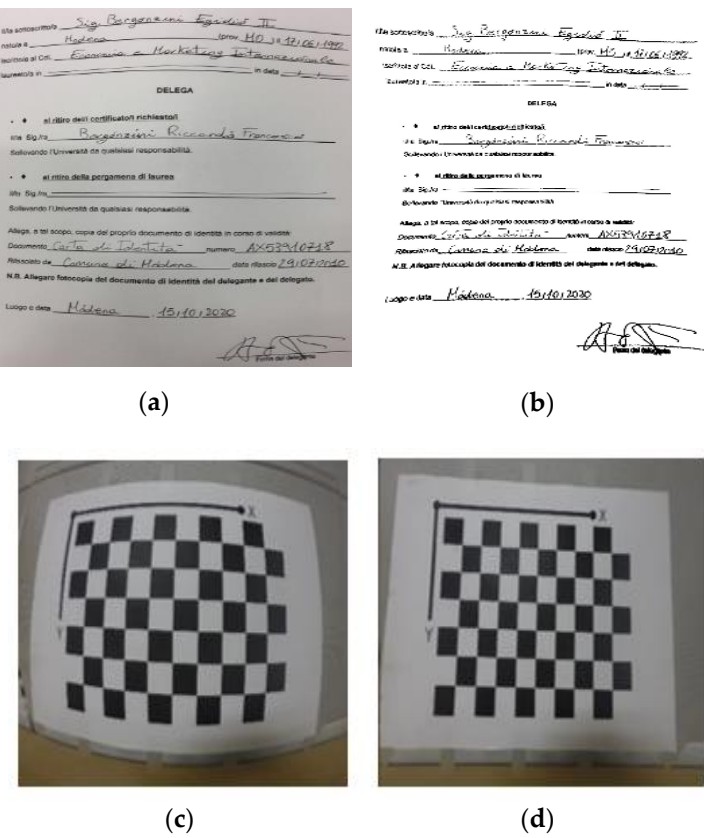

**Figure 10.** De-warping of the image. (**a**) original image. (**b**) de-warped image. (**c**) corresponding checkerboard of the warped document. (**d**) de-warped checkerboard.

We used two different types of testing to analyze and validate the performance of our results: quantitative analysis based on OCR engines (optical character recognition) (error rate Er) and human visual perception. Figures 10–12 show that our proposed method is more readable and easier to comprehend based on visual criteria.

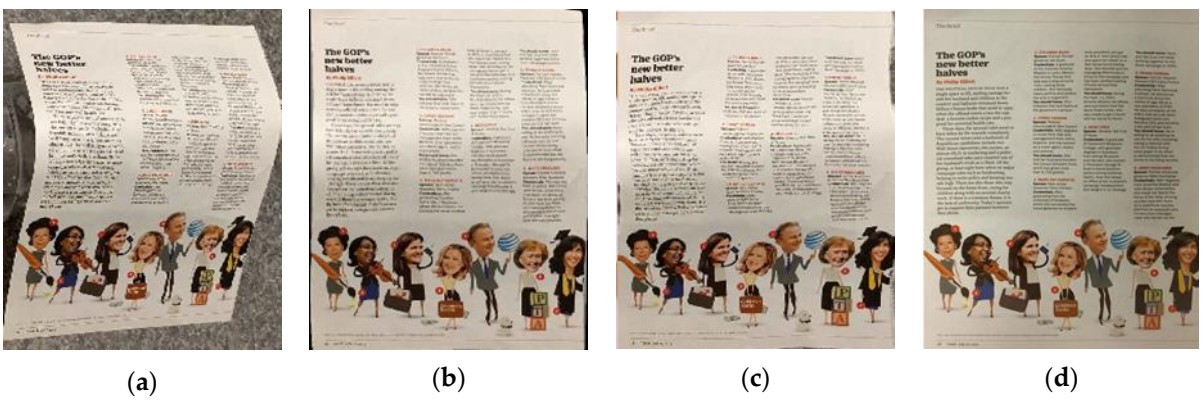

**Figure 11.** Recent de-warping method comparison. (**a**) warped image (**b**) de-warped image by [36] (**c**) de warping image by [37] (**d**) proposed de-warped image.

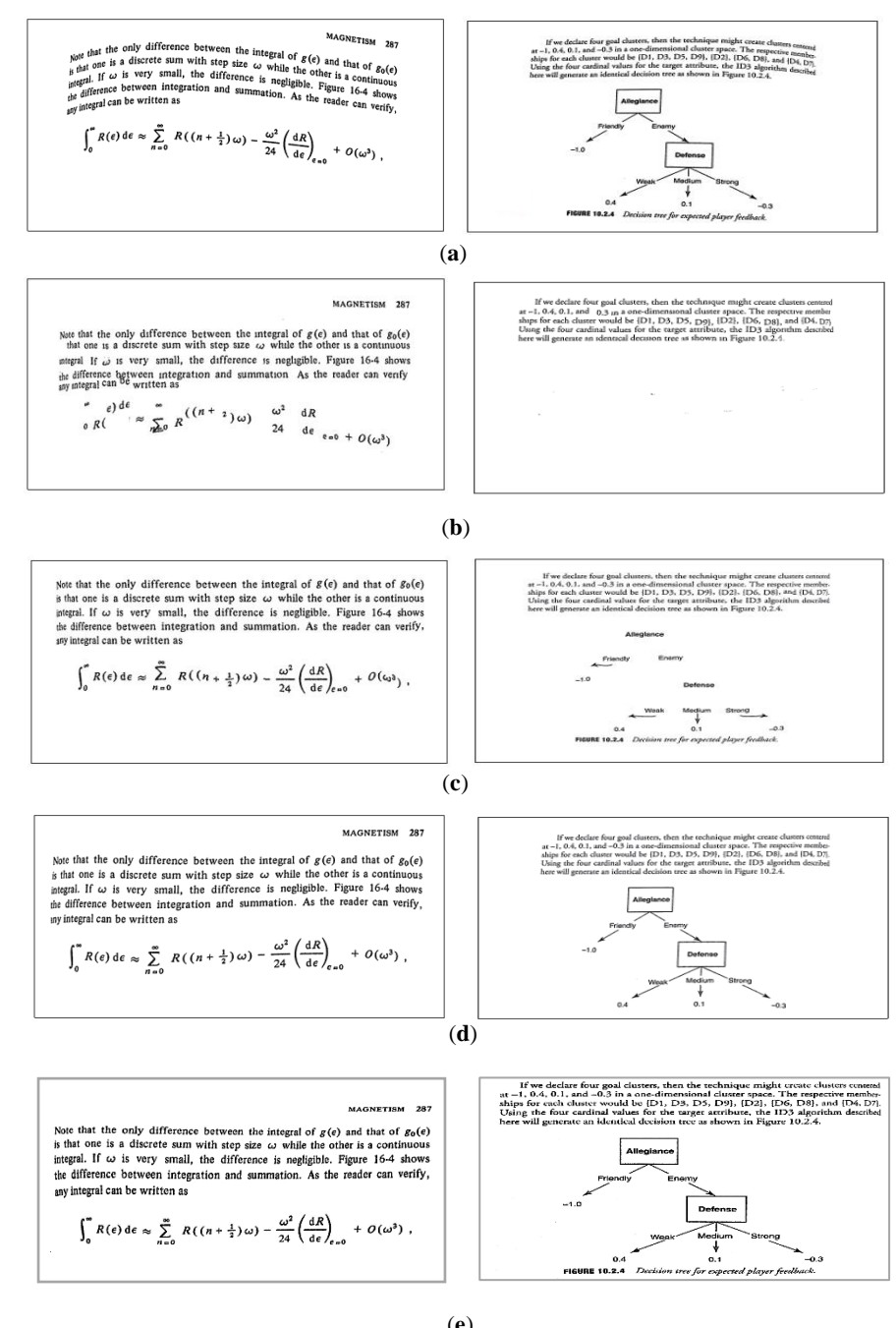

**Figure 12.** The comparison between our approach and other state-of-the-art methods using CBDAR dataset. (**a**) Original document images in CBDAR dataset. (**b**) Method proposed in [14]. (**c**) Method proposed in [4]. (**d**) Method proposed in [13]. (**e**) Our proposed method.

Quantitative analysis based on the OCR engine is used to evaluate and validate the performance of our results, and to test the text region of the proposed results against the state-of-the-art methods. In this work, the ABBYY Fine Reader OCR program was used. The ABBYY program is used to compute the number of characters in the document image before and after the application of a de-warping technique. The software is used to convert image files, scanned documents, and PDF documents into editable formats. The ABBYY application is employed to calculate the number of characters in the document image.

The proposed results are compared to the state-of-the-art methods in [13,36] using OCR performance as a quantitative analysis. The most popular method for evaluation is to

use OCR performance (error rate Er). The OCR error rate (Er) metric is the most common method used for evaluation, and works by computing the ratio incorrect characters to the total number of characters in the document image as shown in Equation (12).

$$\text{Er} = \frac{incorrect\ character}{all\ character} * 100\% \tag{12}$$

Characters which the OCR fails to recognize are called incorrect characters.

Table 1 illustrates the error rate (Er) of our method and the methods in [13,36]. The proposed method achieved a low error rate.

**Table 1.** The performance of OCR (error rate Er).

| | |
|---|---|
| Original image | 40% |
| Approach in [36] | 20% |
| Approach in [13] | 18% |
| Our method | 4% |

The only available document de-warping dataset (CBDAR 2007) was employed to evaluate our method. A total of 102 camera-captured documents are included in this dataset, along with their associated ASCII ground-truths. We compared our method with other state-of-the-art approaches in [4,13,14]. Figure 12 presents two examples from the dataset. The selected two images have one of the most challenges that most of the previous methods fail to solve. Figure 12a is the original warped document image from dataset. Figure 12b shows the de-warped image proposed by [14]. The method in [14] produced bad results in the equation portion of the left image and failed when the right document image contained figures. Figure 12c shows the de-warped image proposed by [4]. The results of the method in [4] are better than the method in [13] in the left image but still failed in the right image due to the figure inside the document. Figure 12d shows the de-warped document image which was proposed by [13]. The results of the recent method in [13] solved the problem of the document that contains figures but removed the title of the figure. Furthermore, it failed to de-warp the equation and text in the left image. Figure 12e illustrates the de-warped image using our method, which produces straight lines in the text part and is efficient in the case of the image that contain figures and equations.

Finally, our method outperforms the other approaches, regardless of the content of the document images, as shown in Table 2 and Figure 12. In the case of figures, equations, and tables of various levels of complexity and poor illumination, the proposed method provides high-quality results.

**Table 2.** The performance of OCR (error rate Er).

| | |
|---|---|
| Original image | 42% |
| Approach in [4] | 25% |
| Approach in [14] | 19% |
| Approach in [13] | 12% |
| Proposed method | 3% |

## 5. Conclusions

In this paper, we present the de-warping of a document image based on a checkerboard calibration pattern. The proposed method recovers the deformed document using a checkerboard calibration pattern based on a camera-calibration algorithm, in spite of the content of the image. Finally, the warping issues are corrected using the radial distortion technique. Based on the CBDAR 2007 dataset, we evaluated the proposed method on a variety of warped document images. When compared to other state-of-the-art methods, the results demonstrated a higher level of quality. The proposed method corrects the warping

issue in document images of various levels of complexity, and is successful in recovering the image of a document that includes figures and tables.

**Author Contributions:** Conceptualization: M.I., M.W. and S.Z.; Methodology: M.I. and M.W.; Formal analysis and investigation: W.S.E., F.S.A. and A.M.Q.; Writing—original draft preparation: M.I., M.W. and S.Z.; Writing—review and editing: W.S.E., F.S.A. and A.M.Q.; Resources: M.I., M.W. and W.S.E.; Visualization: S.Z. and A.M.Q.; Supervision: M.I., S.Z. and W.S.E. All authors have read and agreed to the published version of the manuscript.

**Funding:** This work was supported by Taif University Researchers Supporting Project number (TURSP-2020/347), Taif University, Taif, Saudi Arabia.

**Acknowledgments:** The authors thank Taif University Researchers Supporting Project number (TURSP-2020/347), Taif University, Taif, Saudi Arabia.

**Conflicts of Interest:** The authors declare no conflict of interest.

**Ethical Approval:** This article does not contain any studies with human participants or animals performed by any of the authors.

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
