# Peer review of "An Efficient Method for Document Correction Based on Checkerboard Calibration Pattern"

_applsci, doi:10.3390/app12189014_

Round 1

Reviewer 1 Report

This paper proposes a novel document image de-warping method based on checkboard cal-ibration pattern. It seems that the authors only consider the geometric distortion caused by the lens. Using checkboard is not a natural way to correct the geometry of the text on non-plannar papers. The english of the paper needs to be improved.

Reviewer 2 Report

This paper deals with the problem of document correction based on Check-board calibration pattern. Specifically, a checkboard calibration pattern is utilized to calculate the world and image points, based on which a radial distortion algorithm is used to handle the warping problem. Experimental results demonstrate the effectiveness of the proposed method. However, I have the following concerns:

1.     The problem of document correction based on computer vision has been investigated for decades, and the new challenges faced in this paper should be further specified.

2.     The main contribution of the proposed method is the introduction of a checkboard calibration pattern. The checkboard pattern is manually chosen for further use, which is the main bottleneck of the method.

3.     More recent works in this field should be included and compared.

4.     The writing of the manuscript should be improved, including the quality of the figures and the languages.

5.     The definition of the performance metric should be clearly described, such as incorrect character, and OCR.

Reviewer 3 Report

Authors have proposed method for dewarping of images which may arise due to folded pages of book or for some other reasons. The manuscript is well presented and authors have worked on a recent and practical problem. Overall the manuscript is good , but following points need to be addressed by the authors.

1. There are some instances of wrong usage of English language like gotten on Page 2. It should be changed to got. Authors are encouraged to check their manuscript for such mistakes.

2. On Page 3, authors have mentioned on other hand. However, there is no mention of first hand. Authors are advised to reword it.

3. The start of section 3. Proposed method: Authors should move the text from first line till sixth line upto inaccurate to the last of precious section and then add 

In order to overcome these problems, we have proposed a novel techniqu which is discussed in next section.

This will bring a flow to the section change.

4. Give some gap between figure 10 and it's previous paragraph.

5. Some recent references need to be added. As the authors have mentioned about portable multimedia devices, the following paper should also be cited:

M. Tausif, A. Jain, E. Khan and M. Hasan, "Low Memory Architectures of Fractional Wavelet Filter for Low-Cost Visual Sensors and Wearable Devices," in IEEE Sensors Journal, vol. 20, no. 13, pp. 6863-6871, 1 July1, 2020, doi: 10.1109/JSEN.2019.2930006.

Round 2

Reviewer 2 Report

The authors have partially improved the quality of the manuscript. Although some comments are ignored and a response letter is missing, the work is recommended to the journal.